# Impact of Polymorphisms in the Serotonin Transporter Gene on Oscillatory Dynamics during Inhibition of Planned Movement in Children

**DOI:** 10.3390/brainsci9110311

**Published:** 2019-11-06

**Authors:** Andrey V. Bocharov, Alexander N. Savostyanov, Sergey S. Tamozhnikov, Alexander E. Saprigyn, Ekaterina A. Proshina, Tatiana N. Astakhova, Gennady G. Knyazev

**Affiliations:** 1Laboratory of Psychophysiology of Individual Differences, Institute of Physiology and Basic Medicine, 630117 Novosibirsk, Russia; a.n.savostyanov@physiol.ru (A.N.S.); stam@physiol.ru (S.S.T.); saprigyn@physiol.ru (A.E.S.); proshinaea@physiol.ru (E.A.P.); knyazev@physiol.ru (G.G.K.); 2Humanitarian Institute, Novosibirsk State University, 630090 Novosibirsk, Russia; tastahova95@yandex.ru; 3Laboratory of Psychological Genetics, Institute of Cytology and Genetics of SBRAS, 630090 Novosibirsk, Russia

**Keywords:** stop signal paradigm, alpha oscillations, beta oscillations, serotonin transporter polymorphism, inhibition

## Abstract

The serotonin transporter plays an important role in serotonergic neuromodulation. It is known that polymorphisms in the serotonin transporter gene are linked to the dysregulation of emotions. In the current study, we aimed to investigate the impact of variation in the gene encoding serotonin transporter polymorphism on oscillatory dynamics during inhibition of planned movement in the stop signal paradigm. During performance the stop signal paradigm, electroencephalograms (EEGs) were recorded in 90 healthy Caucasian children (39 girls) from 7 to 12 years. Buccal epithelium probes were taken from all participants. The La, Lg, and S alleles of serotonin transporter-linked polymorphic region (5-HTTLPR) polymorphism were detected in the DNA samples using PCR. LaLa genotype carriers did not differ from LaS\LgS and LgS\LgLg\SS carriers in a number of successful inhibitions of a prepotent response. Carriers of LaLa exhibit higher alpha and beta event-related spectral perturbations (ERSP) in parietal and occipital cortices after the presentation of signal of inhibition of movement than LaS\LgS and LgS\LgLg\SS carriers. Results are consistent with current literature and confirm that S allele carriers are more predisposed to disturbances in cognitive control.

## 1. Introduction

In recent years, a particular polymorphism in the serotonin transporter has attracted a lot of attention from researchers. The serotonin transporter has a great importance in serotonergic neuromodulation, which is strongly linked to mood, emotion, and regulation of cognition [1]. The serotonin transporter facilitates the reuptake of serotonin from the synaptic cleft at the brain synapses. The serotonin gene is located on chromosome 17q12, and encoded by gene SLC6A4. The transcription of gene SLC6A4 is modulated by two polymorphisms, resulting in the long (L) and short (S) variants of the gene [2]. The S allele is linked to a lower level of RNA transcription compared with the L allele. In addition, the L allele contains an A/G single nucleotide polymorphism. The La allele is associated with a higher transcriptional efficiency, whereas Lg allele is connected to lower transcriptional efficiency and is similar to the S allele [3].

Studies have shown that serotonin transporter-linked polymorphic region (5-HTTLPR) polymorphism, specifically the SS genotype, is associated with a higher risk of depression, impulsivity, anxiety, post-traumatic stress disorders, and an increase in neuroticism, and sensitivity to negative emotional cues [2,4,5,6,7,8].

Thus, according to Caspi et al. [4], the risk of depression and post traumatic disorders in patients with these polymorphisms can be observed more frequently when they are put under stress or adverse environmental conditions. Moreover, the development of major depressive disorder under conditions of stress has been shown to occur more frequently in patients who are homozygous for the short (S) allele compared to the long (L) allele [4]. In Bogdan et al. [9] research in a group of 234 children has shown similar results to Caspi et al. [4]. In this experiment, children who were homozygous for the short (S) allele had a higher rate of predisposition to depressive disorders under stress.

Interestingly, impulsive people can be characterized by a reduced ability to control their behavior. A range of studies have shown that carriers of the S allele were more prone to impulsive behavior and had exhibited a higher level of neuroticism, as well as scoring lower on their ability to inhibit certain behaviors [5,10,11]. Other studies failed to show the connection between impulse control disorders and a polymorphism of the 5-HTTLPR gene in a non-clinical sample of subjects [12,13].

Keeping in mind the above findings [2,4,6,7,8,9,10], it could be suggested that carriers of the S allele may have a decreased ability of emotional and behavior regulation. This may be caused by reduced cognitive inhibitory control. Currently, there is no consensus in regard to the effects of the polymorphisms on behavior [12,13]. However, these effects may be observed in the differences in brain activity of LL, LS, and SS genotype carriers.

In studies looking at foundations of neurocognitive emotional disturbances, simple motor control tasks are often used [14]. It has been shown that different kinds of inhibitory control (motor and affective inhibitory control) share a common psychobiological substrate [14,15]. In addition, impaired motor inhibitory control was associated with depression and impulsivity [16,17]. For instance, children with attention deficit hyperactivity disorder (ADHD) took longer to inhibit their initial motor response, and reported disturbances in emotional regulation [18]. According to the study by Carlson and Wang [19], individual differences in inhibitory control were related to the ability of children to regulate their emotions. Moreover, motor and emotional inhibitory control measured in stop-signal tasks and emotion reappraisal tasks were related in adults [14].

It has been shown that carriers of S allele were more predisposed to disturbances in emotional regulation, the risk of posttraumatic stress, depression, and anxiety disorders [20,21,22]. According to Minelli et al. [21], development or presence of emotional disturbances in S allele carriers could lead to inaccurate data on behavioral and brain activity measures gathered from studies involving genetic polymorphisms. It can be assumed that the risk of developing psychopathologies increases with age due to the accumulation of stress and an increase in the number of stressful situations. An advantage of this study was that it would be performed on mentally and physically healthy children.

Our research focused on motor control, which is one of the aspects of cognitive control. The stop signal paradigm is a well-established paradigm for the study of pre-planned movement [23]. This task consists of a pseudorandom mix of go and stop trials. In go trials, subjects have to move quickly and accurately in response to the presentation of the target stimulus. Less frequently, a stop signal is presented during the reaction time (RT) and the participant is instructed to inhibit the pending action. The stop signal paradigm has an advantage compared to other tasks where two signals must be processed in fast sequences, as it assumes there is no interference between the processes responding to the go stimulus and the processes responding to the stop signal stimulus. This is important, because it suggests that response inhibition is not subject to capacity limitations that prevail in other dual task situations [24].

To date, little is known about the effects of 5-HTTLPR polymorphism on brain activity in children. Specifically, there is a lack of literature on how the 5-HTTLPR polymorphism affects neural oscillations during inhibition of planned motor reaction in children.

The aim of the current study is to investigate the effect of the 5-HTTLPR gene polymorphism on oscillatory dynamics during successful inhibition of motor reaction in the stop signal paradigm in healthy children. 

According to the study by Babiloni et al. [25], movement execution and movement observation induced a desynchronization of alpha and beta bands. Whereas an increase of alpha oscillations was related to the inhibition of various cognitive and motor tasks, an increase of the lower beta frequency range was involved in motor inhibition [26,27,28,29]. Based on this, we assume that inhibition of a motor reaction may be accompanied by an increase in the spectral power of alpha (7–13 Hz) and lower beta (13–17 Hz) rhythms. Another expectation is that for carriers of LL genotype, who are linked to exhibiting a higher resistance to the development of disorders of the motivational-affective sphere and higher abilities of cognitive control, such increase in alpha and lower beta rhythms could be higher than in S allele carriers.

## 2. Materials and Methods

### 2.1. Participants

The sample involved 90 healthy Caucasian children (39 girls) (mean age was 9.1; standard deviation was 1.3) aged between 7 and 12 years. Study participants were right-handed and had normal or normalized vision. Informed consent for inclusion was signed by parents of every child before they participated in the study. The research was conducted in accordance with the Declaration of Helsinki. Ethics Committee of the Research Institute of Physiology and Basic Medicine approved the study.

### 2.2. Genotyping

Buccal epithelium probes were taken from all participants. The La, Lg, and S alleles of 5-HTTLPR polymorphism were detected in the DNA samples using polymerase chain reaction (PCR) with primers: 50-gagggactg agctggacaacccac-30 and 50-ggcgttgccgctctgaattgc-30 [2]. 5-HTTLPR polymorphism was detected by agarose-gel electrophoresis. The L and S alleles sizes for 5-HTTLPR were 529 bp and 489 bp, respectively. To identify La and Lg alleles, a digestion of the products of amplification during three hours with MspI endonuclease was made. After the digestion, the sizes of the products were 340, 127, and 62 bp for the La allele, and 174, 166, 127, and 62 bp for the Lg allele.

The triallelic classification was reclassified into a biallelic model by a level of transporter expression: La/La genotype was classified as LL (high level of transcriptional efficiency), La/S and La/Lg genotypes were classified as LS (intermediate level of transcriptional efficiency), and Lg/S, Lg/Lg, and S/S genotypes were classified as SS (low level of transcriptional efficiency) [30].

### 2.3. Stop Signal Paradigm

The stop signal paradigm was realized as a computer game with the presentation of one out of two go stimuli (a rabbit or a tiger). During the game, the subject was motivated to achieve the maximum score. This task was based on the stop signal task for adults described in previous studies [31,32] and was adapted for children.

The images (rabbit or tiger) were presented randomly at the center of the screen (17 cm × 17 cm) during a time period of 500 ms. The images of carrot and meat were at the bottom right and left of the screen, respectively. Participants were asked to choose food (carrot or meat) to feed the rabbit or tiger by pressing a respective button in accordance with the type of go stimulus (a carrot for a rabbit; meat for a tiger).

Prior to the start of the task, the following instruction appeared on the screen. “During the game, a rabbit or a tiger will appear on the screen. You need to choose the right food—carrot for the rabbit (button K) or meat for the tiger (button D) and press the button before the animal disappears. If a “STOP” signal appears on the animal, then nothing should be pressed. If you correctly feed the animal, you will receive a point, and in case of a wrong response (wrong food choice or if action was carried out after the “STOP” signal), the point will be deducted”.

Participants were instructed to press left or right buttons by left or right index fingers, respectively. The response keys were not counterbalanced across subjects.

There were 160 trials. The first 30 trials without stop signals were used in the training session. The practice session consisted of a pseudorandom mix of 96 (approximately 96%) no-stop and 34 (approximately 26%) stop trials. In the stop signal trials, subjects had to refrain from pressing any button if a stop signal (red rectangle containing the word “STOP”, 250 ms in duration) was presented after the go stimulus. Subjects were instructed to respond as fast and as accurately as possible. The interstimulus interval randomly varied between 3.5 and 5.5 s.

As described in the study by Mirabella et al. [33], participants failed to inhibit their movement more frequently when the time interval between go and stop signals increased. Similarly, in this study we computed the time interval between the onset of the go and stop signal stimuli (stop signal delay, SSD) for each participant based on the average RT measured in the first 30 trials. The SSD was 10%, 20%, 70%, and 80% to the average RT. Thus, for instance, if the mean RT in the training section was 600 ms, then four variants of time interval between the go stimulus and stop stimulus (600 × 0.1; 600 × 0.2; 600 × 0.7; 600 × 0.8) would be 60 ms, 120 ms, 420 ms, and 480 ms, respectively.

If a response to the go stimulus was correct, the participant received additional points, and a deduction of points was applied to mistakes and RT later than 750 ms. The trials containing wrong responses were rejected from analysis. Also, the first 30 trials from the training session for the choice reaction task without stop signals were rejected. 

### 2.4. Electroencephalography (EEG) Records

The EEGs were recorded from 64 electrodes referred to Cz. A mid-forehead electrode was the ground. The signals were amplified with a multichannel biosignal amplifier with bandpass 0.1–100 Hz, and continuously digitized at 1000 Hz. The horizontal and vertical electrooculograms were registered simultaneously.

### 2.5. Data Analysis

Artifacts were eliminated by independent components analysis implemented in the EEGLAB toolbox software (Swartz Center for Computational Neuroscience (SCCN) of the University of California San Diego (UCSD), USA) (http://www.sccn.ucsd.edu/eeglab). The time-frequency decomposition was calculated using Morlet wavelets with the number of cycles linearly increasing with frequency beginning at 1.5 cycles and capping at 8 cycles at 40 Hz.

To assess stop signal presentation-evoked changes in spectral power, event-related spectral perturbations (ERSP) were calculated using EEGLAB toolbox. The event-related spectral perturbations show mean logarithm event-locked deviations from baseline-mean power at each frequency [34]. Under the Stop condition, the time interval began at 1250 ms prior to the stop signal presentation and continued for 750 ms; this interval was used as the baseline. The following 1000 ms after stop signal onset were used as the test interval. 

It has been repeatedly shown that in healthy young adults, the time it takes to the stop signal, or stop signal reaction time (SSRT) was about 200 ms [33,35]. In healthy children, the time interval of inhibition (stop signal RT interval) was about 220 ms [36]. In this study we did not compute the SSRT, but we rely on measures taken from the existing literature [33,35,36,37], thus we considered the first 200 ms after the presentation of the stop signal as the SSRT.

ERSP were averaged in 3 frequency ranges (alpha1, 7–10 Hz; alpha2, 10–13 Hz; and beta1, 13–17 Hz), and 12 cortical regions (left frontal: FP1; AF3, F7, F5, F3, FT7, FC3, FC5, FC1, F1; right frontal: FP2, AF4, F8, F6, F4, FT8, FC4, FC6, FC2, F2; middle frontal: AFz, Fz, FCz; left temporal: T7, TP7, TP9; right temporal: T8, TP8, TP10; middle central: Cz; left central parietal: C1, C3, C5, CP1, CP3, CP5; right central parietal: C2, C4, C6, CP2, CP4, CP6; middle central parietal: CPz, Pz; left parietal occipital: P1, P3, P5, PO3, PO7, O1; right parietal occipital: P2, P4, P6, PO4, PO8, O2; middle parietal occipital: POz, Oz). Two cortical region factors, laterality (left, middle, or right region) and cortex (frontal, central, parietal, or occipital cortex) were used in the analysis.

Statistical analysis was performed using the SPSS software package (IBM, NY, USA). A three-way ANCOVA was performed using the following factors: Frequency range (3 levels), laterality (3 levels), and cortex (3 levels) as within-subject factors, polymorphism of 5-HTTLPR (LL, LS, and SS) as a between-subject factor, and age as a covariate. For correction of violations of sphericity the Greenhouse–Geisser correction was used as appropriate.

One-way ANOVAs were performed to reveal significant differences between LL, LS, and SS groups.

## 3. Results

The distribution of genotypes was LL—41.1%, LS—40%, and SS—18.9%. One-way ANOVAs were conducted to reveal possible differences between the genotype groups. Composition of groups LL (*n* = 37, mean age = 9.1, SD = 1.3), LS (*n* = 36, mean age = 9, SD = 1.3), and SS (*n* = 17, mean age = 9, SD = 1.1) did not differ in age (F = 0.27, *df* = 2, *p* = 0.77). Groups of research participants LL (22 male, 15 female), LS (19 male, 17 female), and SS (10 male, 7 female) did not differ in sex (F = 0.18, *df* = 2, *p* = 0.84). 

There were no statistically significant differences between the genotype groups on subscales of emotional symptoms, conduct problems, hyperactivity-inattention, peer problems, and prosocial behavior, which were measured using the Strengths and Difficulties Questionnaire (SDQ) [38,39]. The genotype groups also did not differ in the level of neuroticism, extraversion, disagreeableness, and openness, which were measured using the Inventory of Child Individual Differences (ICID) [40] (Table 1).

The groups LL, LS and SS had the same results for the number of successful inhibitions of motor reactions after the presentation of the stop signal (F = 0.4, *df* = 2, *p* = 0.68). The average RT in “Go” condition was 623 ms, SD = 35. One-way ANOVAs revealed no differences in RT in “Go” condition between the groups LL, LS, and SS (F = 0.18, *df* = 2, *p* = 0.84), as well as no differences in the number of successful “Go” reactions (F = 0.13, *df* = 2, *p* = 0.88) (Table 1).

A three-way ANCOVA was performed using the following factors: frequency range (3 levels), laterality (3 levels), and cortex (3 levels) as within-subject factors; the number of inhibitions of prepotent response as a between-subject factor; age as a covariate. The interaction between the number of inhibitions of the prepotent response after the presentation of the stop signal, cortical area, and laterality was marginally significant (F = 1.4, *df* = 77, *p* = 0.054). In order to understand this interaction, we made a median split of the number of inhibitions of the prepotent response on the stop signal (a median equals 65%). In the current study, the lowest result was 35% from all shown stop signals, and the maximum was 94%. In this study, we used only those EEG epochs in which the participants successfully inhibited their motor reaction (i.e., did not press the button).

In the interval from the beginning till 200 ms after the stop signal presentation, the average spectral power in the range from 7 to 17 Hz did not differ in groups with high and low results in the task (Figure 1).

A three-way ANCOVA was performed using the following factors: Frequency range (3 levels), laterality (3 levels), and cortex (3 levels) as within-subject factors, polymorphism of 5-HTTLPR (LL, LS, and SS) as a between-subject factor, and age as a covariate. A significant effect of subjects with existing polymorphisms in ERSP scores was revealed (F = 3.4, *df* = 2, *p* = 0.038). The participants with LL genotype had a significantly higher increase in spectral power in the frequency range from 7 to 17 Hz during 200 ms after stop signal presentation.

A significant interaction between polymorphisms in 5-HTTLPR, frequency range, cortical area, and laterality has been identified (F = 2.23, *df* = 6.9, *p* = 0.029).

One way ANOVAs between LL, LS, and SS groups revealed the following results.

In alpha1 (7–10 Hz) band ERSP scores were greater for a LL genotype group in occipital cortex (F = 4.28, df = 2, *p* = 0.017) (Figure 2).

In the alpha2 (10–13 Hz) frequency range ERSP scores after the stop signal were higher in the LL genotype in parietal (F = 4.28, *df* = 2, *p* = 0.017) and occipital areas (F = 4.69, *df* = 2, *p* = 0.012) (Figure 3).

In the beta1 (13–17 Hz) frequency range ERSP scores after the stop signal were higher in the LL genotype in parietal (F = 3.99, *df* = 2, *p* = 0.022) and occipital areas (F = 3.72, *df* = 2, *p* = 0.028). In frontal cortex ERSP scores of beta1 band were lower in LS genotype group (F = 3.95, *df* = 2, *p* = 0.023) (Figure 4).

## 4. Discussion

It was initially hypothesized that the alpha and lower beta rhythms could increase during inhibition of planned motor reactions in the LL genotype group. This hypothesis was proved by our results. Thus, our results indicate that the group of healthy LL genotype children tended to have an increase in the spectral power during the inhibition of planned motor reactions in the alpha and lower beta ranges, mainly in the parietal and occipital areas of the cortex.

According to the inhibition–timing hypothesis, an increase of alpha oscillations could be related to a process of inhibitory control [26]. Hummel et al. [41] have shown that inhibitory control of the movement was accompanied by a pronounced synchronization of upper alpha spectral power over the sensorimotor areas. According to Ogrim, Kropotov, and Hestad [42], beta spectral power in healthy children correlated with higher attention level, and an increase of lower beta rhythm was involved in motor inhibition [27,28,29,43]. In general, therefore, it seems that the increase of alpha and lower beta rhythms in response to a suppression signal of a motor reaction may reflect better abilities of cognitive control in carriers of the LL genotype.

The differences in features of oscillatory dynamics during the process of inhibition have been revealed in groups of LL genotype and S allele carriers. However, the polymorphisms in 5-HTTLPR did not impair the subjects’ ability to suppress a prepotent movement. Thus, no differences were revealed in the amount of successful inhibitions of a prepotent response, as well as in a successful reaction to the go stimuli and RT in the “Go” condition between the genotype groups (LL, LS, and SS).

Clark et al. [12] showed that the polymorphisms in 5-HTTLPR did not affect behavioral responses to inhibitions caused by a stop signal, and there was no effect on the “Go” condition. Landro et al. [10] have shown that where a stop signal paradigm was used, an effect of the polymorphism in 5-HTTLPR was observed. However, this effect was revealed only in the measurement of the average RT to a stop signal, which reflects the effectiveness of the subject to be able to inhibit a prepotent reaction. The RT was the slowest in LL genotype carriers, and the quickest in SS genotype. In our research, the measurement of the average time of inhibition was not used; only those EEG epochs, which had successful inhibition of planned reactions when showing a stop signal, were used for the oscillatory dynamics analysis.

Studies that used the stop signal paradigm to determine if there is an effect between the polymorphisms and personal traits demonstrated contradictive results. According to a meta-analysis by Minelli et al. [21], a lack of psychiatric screening of study participants may lead to significant deviations in genetic studies due to the higher incidence of depressive and anxiety disorders in carriers of the S allele. Thus, it has been shown that Caucasoid carriers of SS genotype had higher anxiety related scores; however, in studies which used structured psychiatric screening such associations were not found [21].

Thus, the lack of psychiatric screenings and unknown information on personal traits may lead researchers to make an incorrect link between the polymorphisms and behavioral measures. It can be assumed that the risk of development of psychopathologies increases with age due to the accumulation of stress and an increase in a number of stressful situations. Based on this, carrying out research on children becomes an advantage. In addition, it is known that all children that participated in this study were mentally and physically healthy, and were not registered with a psychiatrist.

A limitation of this study was a lack of data on anxiety scores in the groups LL, LS, SS. According to Muris, Meesters, and van den Berg, emotional symptoms of SDQ are correlated to symptoms of anxiety and depression [44]. In this study, the screening of personality traits and problems of the subjects showed no difference between LL, LS, and SS groups.

Therefore, the results of this study agree with the majority of studies on the behavioral measures of LL, LS, and SS carriers in the context of the absence of the effect of polymorhisms in 5-HTTLPR on behavior.

However, differences were revealed on a level of oscillatory dynamics of the brain during inhibition of planned motor actions. In our opinion, they were connected with a higher cognitive potential to control the behavior in carriers of LL genotype. In S allele carriers, a lower alpha and beta spectral power during the inhibition of a planned action may reflect a decrease of a cognitive potential to control the behavior. This may contribute to the development of impairment in the ability to control the behavior under adverse and stressful conditions.

## 5. Conclusions

Healthy children that are carriers of the low expressive 5-HTTLPR polymorphisms (LS and S) did not differ from LL carriers in the number of inhibitions to a prepotent response. However, differences concerning oscillatory dynamics accompanied with inhibition of a prepotent movement were revealed. LS and SS genotype carriers exhibited a lower alpha and beta1 spectral power after the presentation of a signal to inhibit movement, whereas LL carriers showed a more pronounced increase of alpha and beta1 spectral power. These results may be associated to a reduced potential for cognitive control in carriers of the S allele, which may be a predisposing factor in the development of problems related to the regulation of emotions and behavior.

## 6. Limitations 

In this study the SSRT was not measured. Authors relied on measures taken from the existing literature [33,35,36,37] that the first 200 ms after the presentation of the stop-signals were considered as the SSRT.

## Figures and Tables

**Figure 1 brainsci-09-00311-f001:**
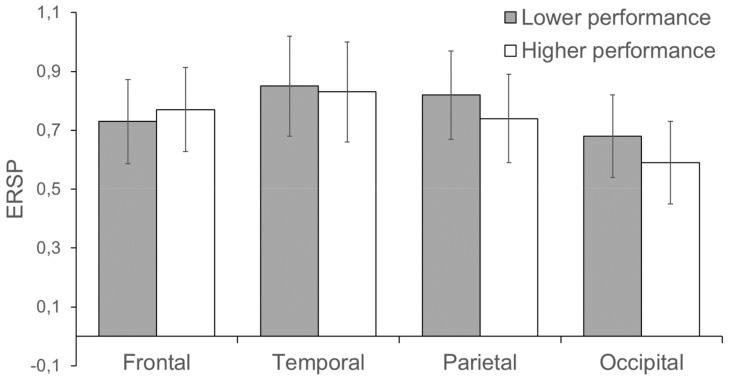
Averaged event-related spectral perturbations (ERSP) scores in the range from 7 to 17 Hz in the time interval from the beginning till 200 ms after the stop signal presentation in groups with performance lower than a median (gray color) and higher performance above the median (white color). Vertical lines reflect the standard error.

**Figure 2 brainsci-09-00311-f002:**
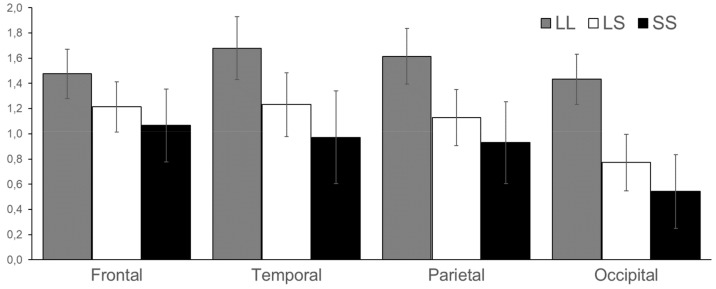
ERSP of alpha1 (7–10 Hz) band in frontal, temporal, parietal, and occipital cortices in LL, LS, and SS groups. Vertical lines reflect the standard error.

**Figure 3 brainsci-09-00311-f003:**
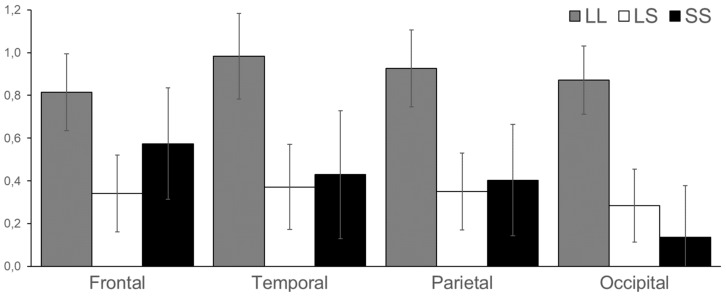
ERSP alpha2 (10–13 Hz) band in frontal, temporal, parietal, and occipital cortices in LL, LS, and SS groups. Vertical lines reflect the standard error.

**Figure 4 brainsci-09-00311-f004:**
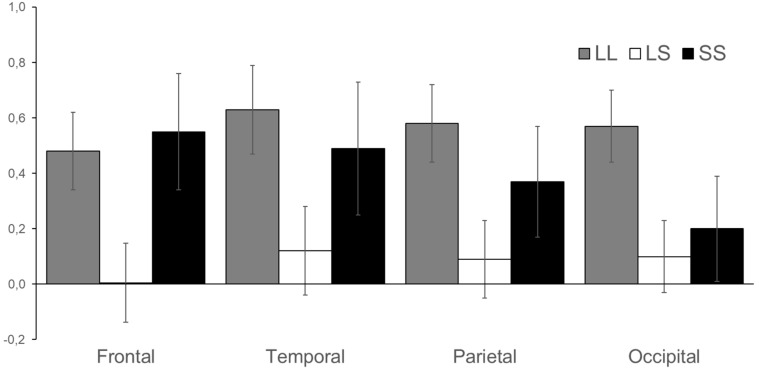
ERSP of beta1 (13–17 Hz) frequency band in frontal, temporal, parietal, and occipital cortices in LL, LS, and SS groups. Vertical lines reflect the standard error.

**Table 1 brainsci-09-00311-t001:** Psychometric data for genotype groups. Values represent mean and standard deviation (mean ± SD). The *p*-values refer to the results from one-way ANOVAs.

	LL (*n* = 37)	LS (*n* = 36)	SS (*n* = 17)	*p* Value
Age	9.1 ± 1.3	9 ± 1.3	9 ± 1.1	0.77
Emotional symptoms	1.16 ± 1.3	1.58 ± 1.57	1.53 ± 1.38	0.42
Conduct problems	1.16 ± 1.19	1.33 ± 1.27	1.76 ± 1.35	0.26
Hyperactivity-inattention	4.2 ± 2.1	3.78 ± 2.4	4.71 ± 2.7	0.39
Peer problems	1.62 ± 1.64	1.97 ± 1.5	1.76 ± 1.44	0.26
Prosocial behavior	7.86 ± 1.62	7.86 ± 1.62	7.29 ± 2.14	0.48
Neuroticism	3.3 ± 0.79	3.34 ± 0.96	3.35 ± 0.82	0.76
Extraversion	4.94 ± 0.64	4.93 ± 0.87	4.86 ± 0.78	0.93
Disagreeableness	2.99 ± 0.67	2.85 ± 0.85	3.37 ± 0.79	0.08
Openness	4.7 ± 0.88	4.85 ± 0.67	4.72 ± 0.72	0.69
Successful “Stop”	67% ± 14%	64% ± 13%	67% ± 13%	0.68
Successful “Go”	65% ± 27%	68% ± 23%	67% ± 25%	0.88
“Go” reaction time	621 ± 29	626 ± 38	618 ± 34	0.84

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
