# Peer review of "Impact of Polymorphisms in the Serotonin Transporter Gene on Oscillatory Dynamics during Inhibition of Planned Movement in Children"

_brainsci, 2019, doi:10.3390/brainsci9110311_

Round 1

Reviewer 1 Report

The paper by Bocharov et al has some potentiality as it aims to describe the relationship between some genetic features of individuals, i.e. the serotonin transporter polymorphism, and the ability to stop a pre-planned movement (i.e. one crucial executive functions). Authors recorded EEG activity during the performance of a stop signal task in healthy subject with different genotypes. Bocharov et al concluded that inhibitory control is decreased in S – allele carriers. However, in my opinion, this conclusion is hampered by a number of potential confounds which are listened below.

Major points

First of all, let a native English speaker revise the paper, and mainly the Result and Discussion sections.

INTRODUCTION

Lines 61-65 First, the stop-signal task is a gold standard for studying action cancellation, whereas the go/no go task is the gold standard for studying action restraint (Schachar et al., 2007). But there are also other forms of inhibitory control such as those that measure interference inhibition, i.e., the ability to resolve response conflict due to irrelevant but incompatible (interfering) stimulus features such as the Simon task, the Eriksen flanker task and the Stroop task. You must specify why you are willing to use this task. Second, you must refer to some previous work; in my opinion, the best is Logan et al. 1984. Third, your explanation of the task is, at best, unclear. The stop-signal task consists of a pseudorandom mix of no-stop and stop trials. Not, as it seems from your sentences, by two different tasks. Lines 71-77 You must make clear that the alpha and the beta rhythms are not explicitly involved in inhibitory control, but more generally are involved in movement execution and movement observation as well (Babiloni et al. 2016). Thus you can find a modulation of those rhythms when an action is inhibited, but changes in those bands are not linked only to inhibitory control.

METHODS

Paragraph 2.3 The description of the stop task is incomplete. How many go and stop trials are in one block of the stop-signal task? What are the percentages? How many blocks? Did you counterbalance the response key across participants? Lines 112-116 This is the procedure used by Mirabella et al. 2006; please refer to that work. Besides, you must provide the inhibition functions and compute the stop signal reaction times (SSRT). The formers allow to see the performance of the individuals or, if normalized, the average performance of the population (see Mirabella et al. 2006). The latter is a crucial measure of reactive inhibitory control. You must absolutely know the length of the SSRT in order to interpret the meaning of your EEG recordings. Only changes occurring between the presentation of the stop signal and the end of the SSRT can be related to the process of inhibition. In the worst case, you have to refer to some previous work in which the SSRT was estimated (but this is not the best solution). Line 135 Related to what I was saying above, a period of 1000 ms after the stop signal presentation is by far too long. It has been repeatedly shown that in healthy young adults, the length of the SSRT is about 200 ms (e.g., Mirabella et al. 2006; Mirabella et al. 2009). In healthy children, the SSRT is a bit longer, being around 220 ms (Mancini et al. 2018). You must refer to those values and somehow justify why you used such a long time window. By reading the results, it seems that you split this time windows into epochs of 200 ms. You must describe the rationale for this choice…which could be the estimated length of the SSRT…but you have to refer to some previous work.

RESULTS

The way you presented the statistical analyses is extremely puzzling. In the methods, you seem to declare that you used a four-way ANCOVA (as you have the age as a covariate). However, at lines 163, 168, it seems you present the data as if you would have done ONE-way-ANOVAs. Please, always specify the analysis you have implemented as well as the factors of the ANOVA. Report mean and SD values in a Table, right now, it is tough to catch the values of the variables. Lines 161-168 How did you compute the number of successful inhibitions? Did you collapse all the mistakes done at all different stop signal delays? In any case, you must compare the inhibition functions across the three groups. You must provide a figure showing the average of successful go-trials and the corresponding mean RTs. Line 174 Did you mean “participants successfully inhibited their action”? Suppressing an action is entirely different concerning “slowing down their motor reaction.” Lines 175-until the end. Here there are some very obscure passages. Were the ERPSs computed as an average across electrodes? How did you choose those electrodes? Why did not you distinguish between the different genotypes? Why in figure 3, there is just one ERPS for each genotype? Is this an average across all electrodes? It would not make sense. You have to show ERPS changes across all the marco-regions you have selected. The same applies to figure 4. How alpha 1 and 2 are defined? Why the figure for alpha 1 band (which has not been defined) is not shown? I would strongly suggest to remove figure 3 and 4 and to show only figure 5 and 6 plus the figure for the alpha 1 band. In any case, Results must be extensively reorganized.

DISCUSSION

Here there are too many passages that, in my opinion, are entirely arbitrary (the link between emotion and inhibitory control, a confound between valence and arousal). I will have to re-read this section after the many corrections that must be done in the other sections. Here I will provide just one more comment Lines 236-238 This is not grounded in the literature. It is very well known that the inhibitory network goes well beyond the prefrontal cortex (e.g. Mirabella 2014), involving both cortical areas (e.g. the PMd Mirabella et al 2011; Mattia et al 2013; the M1 Mattia et al 2012; Coxon 2006) as well as subcortical regions (as the STN Mirabella et al. 2012, 2013; Van den Wildenberg et al. 2006; van Wouwe et al. 2017). Please rewrite these sentences following the existing literature.

Minor points

INTRODUCTION

Please, report the frequency bands you are referring to, i.e. what you mean by alpha band and lower beta band. In the literature the frequency ranges are often different.

METHODS

Line 110 Why did you have such a long ITI’

RESULTS

Please, report mean and SD as mean ±SD. Line 164 (and so forth) Please, change “time reaction” with “reaction time” and use the acronym RT. Please, make a multi-panel figure representing the ERSP in all time epochs.

References

Babiloni C, Del Percio C, Vecchio F, Sebastiano F, Di Gennaro G, Quarato PP,Morace R, Pavone L, Soricelli A, Noce G, Esposito V, Rossini PM, Gallese V, Mirabella G  (2016) Alpha, beta and gamma electrocorticographic rhythms in somatosensory, motor, premotor and prefrontal cortical areas differ in movement execution and observation in humans. Clin Neurophysiol. 127:641-54

Logan, G.D., Cowan, W.B., and Davis, K.A. (1984). On the ability to inhibit simple and choice reaction time responses: a model and a method. J Exp Psychol Hum Percept Perform 10, 276-291.

Mancini C, Cardona F, Baglioni V, Panunzi S, Pantano P, Suppa A, Mirabella G (2018) Inhibition is impaired in children with obsessive-compulsive symptoms but not in those with tics. Movement disorder. 33(6):950-959.

Mattia M, Pani P, Mirabella G, Costa S, Del Giudice P, Ferraina S. Heterogeneous attractor cell assemblies for motor planning in premotor cortex. (2013) J Neurosci 33:11155-68.

Mattia M, Spadacenta S, Pavone L, Quarato P, Esposito V, Sparano A, Sebastiano F, Di Gennaro G, Morace R, Cantore G and Mirabella G (2012). Stop-Event-Related Potentials from intracranial electrodes reveal a key role of premotor and motor cortices in stopping ongoing movements. Front. NeuroenG 5:12.

Mirabella G (2014) Should I stay or should I go? Conceptual underpinnings of goal-directed actions. Front SystNeurosci. 8:206.

Mirabella G, Iaconelli S, Modugno N, Giannini G, Lena F, Cantore G (2013) Stimulation of subthalamic nuclei restores a near normal planning strategy in Parkinson's patients. PLoS One. 8(5):e62793.

Mirabella G, Iaconelli S, Romanelli P, Modugno N, Lena F, Manfredi M, Cantore G (2012) Deep Brain Stimulation of Subthalamic Nuclei Affects Arm Response Inhibition In Parkinson's Patients. Cereb Cortex 22:1124-323.

Mirabella G, Pani P, Ferraina S (2009) The presence of visual gap affects the duration of stopping process. Experimental Brain Research 192:199-209

Mirabella G, Pani P, Ferraina S. (2011) Neural correlates of cognitive control of reaching movements in the dorsal premotor cortex of rhesus monkeys. J Neurophysiol. 106:1454-66.

Mirabella G, Pani P, Parè M, Ferraina S (2006). Inhibitory Control of Reaching Movements in Humans. Experimental Brain Research. 174:240-255.

Schachar, R., Logan, G.D., Robaey, P., Chen, S., Ickowicz, A., Barr, C., 2007. Restraint and cancellation: multiple inhibition deficits in attention deficit hyperactivity disorder. J. Abnorm. Child Psychol. 35, 229–238.

van den Wildenberg WP, van Boxtel GJ, van der Molen MW, Bosch DA, Speelman JD, Brunia CH. 2006. Stimulation of the subthalamic region facilitates the selection and inhibition of motor responses in Parkinson's disease. J Cogn Neurosci. 18:626-636

van Wouwe, N.C., Pallavaram, S., Phibbs, F.T., Martinez-Ramirez, D., Neimat, J.S., Dawant, B.M., D'Haese, P.F., Kanoff, K.E., van den Wildenberg, W.P.M., Okun, M.S., et al. (2017). Focused stimulation of dorsal subthalamic nucleus improves reactive inhibitory control of action impulses. Neuropsychologia 99, 37-47

Author Response

The paper by Bocharov et al has some potentiality as it aims to describe the relationship between some genetic features of individuals, i.e. the serotonin transporter polymorphism, and the ability to stop a pre-planned movement (i.e. one crucial executive functions). Authors recorded EEG activity during the performance of a stop signal task in healthy subject with different genotypes. Bocharov et al concluded that inhibitory control is decreased in S – allele carriers. However, in my opinion, this conclusion is hampered by a number of potential confounds which are listened below.

Response: We would like to thank the reviewer for insightful and important comments on our manuscript. We changed the manuscript according to comments of the reviewers and most of changes are shown in red colour.

Major points

First of all, let a native English speaker revise the paper, and mainly the Result and Discussion sections.

Response: Native English speaker revised the paper.

INTRODUCTION

Lines 61-65 First, the stop-signal task is a gold standard for studying action cancellation, whereas the go/no go task is the gold standard for studying action restraint (Schachar et al., 2007). But there are also other forms of inhibitory control such as those that measure interference inhibition, i.e., the ability to resolve response conflict due to irrelevant but incompatible (interfering) stimulus features such as the Simon task, the Eriksen flanker task and the Stroop task. You must specify why you are willing to use this task. Second, you must refer to some previous work; in my opinion, the best is Logan et al. 1984.

Third, your explanation of the task is, at best, unclear. The stop-signal task consists of a pseudorandom mix of no-stop and stop trials. Not, as it seems from your sentences, by two different tasks.

Response: Thank you for the comments. We added the following text in the manuscript:

«The stop-signal task consists of a pseudorandom mix of target and stop trials. The subject is asked to carry out a set movement quickly and accurately in response to the presentation of the target stimulus. The stop signal task allows to measure response inhibition when the participant is asked to inhibit a planned motor action upon the presentation of a stop signal. The stop signal paradigm has an advantage compared to other tasks where two signals must be processed in fast sequences, as it assumes there is no interference between the processes responding to the target stimulus and the processes responding to the stop signal stimulus. This is important because it suggests that response inhibition is not subject to capacity limitations that prevail in other dual task situations [Logan 1984].»

Lines 71-77 You must make clear that the alpha and the beta rhythms are not explicitly involved in inhibitory control, but more generally are involved in movement execution and movement observation as well (Babiloni et al. 2016). Thus you can find a modulation of those rhythms when an action is inhibited, but changes in those bands are not linked only to inhibitory control.

Response: We referred to Babiloni et al. 2016 and changed text:

«According to the study by Babiloni et al. (2016) movement execution and movement observation induced a desynchronization of alpha and beta bands. Whereas an increase of alpha oscillations was related with inhibition in various cognitive and motor tasks, an increase of the lower beta frequency range was involved in motor inhibition (Klimesch, Sauseng, Hanslmayr, 2007; Liebrand et al., 2017; Pfurtscheller, Da Silva, 1999; Swann et al., 2009).»

References:

Babiloni, C., Del Percio, C., Vecchio, F., Sebastiano, F., Di Gennaro, G., Quarato, P. P., ... Esposito, V. Alpha, beta and gamma electrocorticographic rhythms in somatosensory, motor, premotor and prefrontal cortical areas differ in movement execution and observation in humans. Clinical Neurophysiology, 2016, 127(1), 641-654.

Klimesch, W., Sauseng, P., Hanslmayr, S. EEG alpha oscillations: the inhibition–timing hypothesis. Brain research reviews, 2007, 53(1), 63-88.

Liebrand, M., Pein, I., Tzvi, E., Krämer, U. M. Temporal Dynamics of Proactive and Reactive Motor Inhibition. Frontiers in human neuroscience, 2017, 11, 204.

Pfurtscheller, G., Da Silva, F. L. Event-related EEG/MEG synchronization and desynchronization: basic principles. Clinical neurophysiology, 1999, 110(11), 1842-1857.

Swann, N., Tandon, N., Canolty, R., Ellmore, T. M., McEvoy, L. K., Dreyer, S., ...Aron, A. R. Intracranial EEG reveals a time-and frequency-specific role for the right inferior frontal gyrus and primary motor cortex in stopping initiated responses. Journal of Neuroscience, 2009, 29(40), 12675-12685.

METHODS

Paragraph 2.3 The description of the stop task is incomplete. How many go and stop trials are in one block of the stop-signal task?

What are the percentages? How many blocks?

Response: We changed text according to the сomments and added the following sentences:

«There were 160 trials. The first 30 trials without stop-signals were used in the training session. The practice session consisted of a pseudorandom mix of 96 (approximately 96%) no-stop and 34 (approximately 26%) stop trials.»

Did you counterbalance the response key across participants?

Response: The response key did not counterbalance across subjects. We added this sentence into the text.

Lines 112-116 This is the procedure used by Mirabella et al. 2006; please refer to that work.

Response: We referred to the the study Mirabella et al. 2006. We added the following sentence: «As described in the study by Mirabella et al. (2006), participants more frequently failed to inhibit their movement when the time interval between target and stop signals increased. Also, in this study we calculated the time interval between the onset of the target and stop-signal stimuli individually.»

Besides, you must provide the inhibition functions and compute the stop signal reaction times (SSRT). The formers allow to see the performance of the individuals or, if normalized, the average performance of the population (see Mirabella et al. 2006). The latter is a crucial measure of reactive inhibitory control. You must absolutely know the length of the SSRT in order to interpret the meaning of your EEG recordings. Only changes occurring between the presentation of the stop signal and the end of the SSRT can be related to the process of inhibition. In the worst case, you have to refer to some previous work in which the SSRT was estimated (but this is not the best solution).

Response: in this study we did not measure the stop signal reaction times.

We referred to studies Mirabella et al. 2006; Mirabella et al. 2009 and Mancini et al. 2018.

We added the following sentence: «It has been repeatedly shown that in healthy young adults, the length of time interval of inhibition after presentation of stop signal was about 200 ms (Mirabella et al. 2006; Mirabella et al. 2009). In healthy children, the time interval of inhibition (stop signal reaction time interval) was around 220 ms (Mancini et al. 2018). In this study we used first 200 ms after presentation of stop signal as the time interval which coincided with process of inhibition.»

Line 135 Related to what I was saying above, a period of 1000 ms after the stop signal presentation is by far too long.

Response: we used 1000 ms test interval only for calculating ERSP measures in each time-frequency point. After revision we did new analysis where we used ERSP scores averaged in time interval during 200 ms after stop signal presentation (instead of three time interval: from 0 to 200, from 200 to 400, from 400 to 600 in previous analysis). We did not use time interval from 201 to 1000 ms in the analysis.

In addition, oscillatory responses which accompanying some activities (presentation of stimuli or behavioral responses) usually last longer than their manifestation. We use such long interval to see the end of oscillatory responses after presentation of stimuli or behavioral reactions. Also, we used it for calculating ERSPs. We needed to find 1 second baseline interval when subject’s brain was at the rest state.

It has been repeatedly shown that in healthy young adults, the length of the SSRT is about 200 ms (e.g., Mirabella et al. 2006; Mirabella et al. 2009). In healthy children, the SSRT is a bit longer, being around 220 ms (Mancini et al. 2018). You must refer to those values and somehow justify why you used such a long time window. By reading the results, it seems that you split this time windows into epochs of 200 ms. You must describe the rationale for this choice…which could be the estimated length of the SSRT…but you have to refer to some previous work.

Response: we referred to Mirabella et al. 2006; Mirabella et al. 2009 and Mancini et al. 2018.

We added the following sentence:

«It has been repeatedly shown that in healthy young adults, the length of time interval of inhibition after presentation of stop signal was about 200 ms (Mirabella et al. 2006; Mirabella et al. 2009). In healthy children, the time interval of inhibition (stop signal reaction time interval) was around 220 ms (Mancini et al. 2018). In this study we used first 200 ms after presentation of stop signal as the time interval which coincided with process of inhibition».

RESULTS

The way you presented the statistical analyses is extremely puzzling. In the methods, you seem to declare that you used a four-way ANCOVA (as you have the age as a covariate). However, at lines 163, 168, it seems you present the data as if you would have done ONE-way-ANOVAs. Please, always specify the analysis you have implemented as well as the factors of the ANOVA.

 Report mean and SD values in a Table, right now, it is tough to catch the values of the variables.

Response: We specified the analysis and the factors of the ANOVA according to the comments. Also, we reported mean and SD values in Table1.

 Lines 161-168 How did you compute the number of successful inhibitions? Did you collapse all the mistakes done at all different stop signal delays? In any case, you must compare the inhibition functions across the three groups.

You must provide a figure showing the average of successful go-trials and the corresponding mean RTs.

Response: We collapsed all the mistakes done at all different stop signal delays. We tried to build figures, but the bars in the figures were located on the same level (they looked the same height). It does not make sense to show these figures. We provided all results in Table 1.

Line 174 Did you mean “participants successfully inhibited their action”? Suppressing an action is entirely different concerning “slowing down their motor reaction.”

Response: Thank you for indicating this mistake we changed it to «participants successfully inhibited their motor reaction».

Lines 175-until the end. Here there are some very obscure passages.

Were the ERPSs computed as an average across electrodes? How did you choose those electrodes? Why did not you distinguish between the different genotypes?

Response: New results are similar previous results. But there were only one marginally significant interaction between 3 factors (number of inhibitions Х cortical area X laterality). Also, we added the following sentence in the article:

«The interaction between the number of inhibitions of prepotent response after the presentation of the stop signal, cortical area and laterality was marginally significant (F = 1.4, df = 77, p = 0.054).»

There were no any significant interactions of the number of inhibitions with the factor frequency band and the factor genotype group. In this case we should not distinguish between frequency bands and the different genotypes.

Why in figure 3, there is just one ERPS for each genotype? Is this an average across all electrodes? It would not make sense. You have to show ERPS changes across all the marco-regions you have selected. The same applies to figure 4. How alpha 1 and 2 are defined? Why the figure for alpha 1 band (which has not been defined) is not shown? I would strongly suggest to remove figure 3 and 4 and to show only figure 5 and 6 plus the figure for the alpha 1 band. In any case, Results must be extensively reorganized.

Response: We reorganized new results according to the comments.

DISCUSSION

Here there are too many passages that, in my opinion, are entirely arbitrary (the link between emotion and inhibitory control, a confound between valence and arousal). I will have to re-read this section after the many corrections that must be done in the other sections. Here I will provide just one more comment Lines 236-238 This is not grounded in the literature. It is very well known that the inhibitory network goes well beyond the prefrontal cortex (e.g. Mirabella 2014), involving both cortical areas (e.g. the PMd Mirabella et al 2011; Mattia et al 2013; the M1 Mattia et al 2012; Coxon 2006) as well as subcortical regions (as the STN Mirabella et al. 2012, 2013; Van den Wildenberg et al. 2006; van Wouwe et al. 2017). Please rewrite these sentences following the existing literature.

Response: We rewrite discussion section according to the new results.

Minor points

INTRODUCTION

Please, report the frequency bands you are referring to, i.e. what you mean by alpha band and lower beta band. In the literature the frequency ranges are often different.

Response: We reported the frequency bands in introduction section.

METHODS

Line 110 Why did you have such a long ITI’

Response: Oscillatory responses which accompanied behavioral responses last longer than their behavioral manifestation. We used such long ITI interval, because for calculating ERSP response we needed to get 1 second baseline interval when subject’s brain was at the rest state.

RESULTS

Please, report mean and SD as mean ±SD. Line 164 (and so forth)

Response: We reported mean and SD as mean ±SD in Table 1.

Please, change “time reaction” with “reaction time” and use the acronym RT.

Response: We changed it and used the acronym RT.

Please, make a multi-panel figure representing the ERSP in all time epochs.

Response: After revision we used only one time interval.

Reviewer 2 Report

In a current study, the EEG was used to assess the relationship between 5-HTTLPR polymorphism and the ability to control behavior in the settings of the stop-signal task. Authors expected to observe the increase in alpha-1, alpha-2 and beta EEG power spectra associated with canceling of planned action for carriers of the serotonin transporter polymorphism with long alleles (LL) as compared with a carrier of short ones (LS and SS). It was expected that measured the EEG activity at the alpha and beta bands could be used as a marker of inhibitory control effectiveness. As a result, three groups of children (7-12 years)with three different types of polymorphism of 5-HTTLPR (i.e., LL, LS, SS) were investigated. It was revealed that children with LL type of 5-HTTLPR polymorphism are characterized by a more considerable increase in alpha-2 and a smaller decrease in beta activity.

Although the study can be considered as an interesting one, it has several drawbacks that should be addressed in a revision. 

Introduction

In the beginning, the authors claimed that serotonin transporter has high importance for serotonin neuromodulation and regulation of emotions. Further, it was stressed that those persons who have the polymorphism of serotonin transporter with short alleles are prone to be more impulsive. Which makes them less efficient in ability to control behavior in general and cognitive control in particular. 

However, from this logic, it is not clear why authors decided to investigate the motor control task (signal stop-task) instead, for example, the task for emotional regulation and some other activity associated with impulsivity or emotional processing. I think that it should be more specifically explained how exactly emotional regulation and inhibitory control are interrelated both in terms of theoretical and experimental description (based on the EEG data, for instance). In the present variant of introduction, the utilization of the stop-signal task looks odd, and the more precise underpinning of its relation to emotional regulation is needed. 

Authors also should explain why they decided to study children but not adults.

Not always references to the relevant literature are provided. For instance: 

"The serotonin transporter has a great importance in serotonergic neuromodulation, which is strongly linked to the emotion regulation". "In studies of inhibition of motor reactions the stop-signal paradigm was well established".

Methods

The authors should specify how they motivated children to perform the task of getting the maximum score.  

The description of the experimental task is not always unambiguous. Providing the figure with the scheme of the probe with timing and recording details will substantially improve the manuscript. Some information regarding the task is missed: the total number of trials, the way of informing subjects about their performance (deduction and receiving "additional point"). 

The authors should provide an example of how the time interval between the target stimulus and stop-stimulus was calculated. 

The package used for statistical analysis should be specified.

Results

Results section suffers from several drawbacks mainly associated with some degree of disagreement with the Methods section. Presented results revealed in many variants of analysis which wasn't introduced in the methods. More specifically, the authors should justify a number of issues:

The usage of particular subscales for psychological assessment. For what purposes they were used.; The usage of different ANOVAs; The selection of the 7-17 Hz frequency range of EEG activity, especially taking into account that alpha and beta demonstrated oppositely directed effects;

 Without of abovementioned explanation, it is unclear how the performed types of analysis were motivated. Not all of these results were discussed in the rest of the text.

Thus the "results" section should be aligned with "methods" in terms of methods used and the analysis applied.  

Discussion

The discussion regarding the impact of age to the observed phenomenon, i.e., comparison between similar effects demonstrated for the adults, will strengthen the manuscript. 

The link between emotional regulation, cognitive control, and inhibitory control of motor activity should be thoroughly discussed.

Finally, it is not clear why the authors did not try to correlate the psychometric assessment and revealed changes in EEG activity. 

The text of the manuscript contains several lexical mistakes.

Author Response

Introduction

In the beginning, the authors claimed that serotonin transporter has high importance for serotonin neuromodulation and regulation of emotions. Further, it was stressed that those persons who have the polymorphism of serotonin transporter with short alleles are prone to be more impulsive. Which makes them less efficient in ability to control behavior in general and cognitive control in particular.

However, from this logic, it is not clear why authors decided to investigate the motor control task (signal stop-task) instead, for example, the task for emotional regulation and some other activity associated with impulsivity or emotional processing. I think that it should be more specifically explained how exactly emotional regulation and inhibitory control are interrelated both in terms of theoretical and experimental description (based on the EEG data, for instance). In the present variant of introduction, the utilization of the stop-signal task looks odd, and the more precise underpinning of its relation to emotional regulation is needed.

Response:

We would like to thank the reviewer for insightful and important comments on our manuscript. We changed the manuscript according to comments of the reviewers and most of changes are shown in red colour.

Investigators of the neurocognitive underpinnings of emotional disorders often use simple motor inhibitory control tasks, with the assumption that these tasks share psychological and neural components with the more complex processes of impulsivity and affective regulation (Tabibnia et al., 2011).

Also, we added the following text to the manuscript:

«In studies looking at foundations of neurocognitive emotional disturbances, simple motor control tasks are often used (Tabibnia et al., 2011). It has been shown that impaired motor inhibitory control was associated with depression and impulsivity (Palmwood, Krompinger, Simons, 2017; Chamberlain, Sahakian, 2007). Children with ADHD had slower reaction time to signal the inhibition along with disturbances in emotional regulation (Nigg, 2001). It has been shown that different kinds of inhibitory control (motor and affective inhibitory control) share a common psychobiological substrate (Cohen, Lieberman, 2010; Tabibnia et al., 2011). According to the study by Carlson and Wang (2007) individual differences in inhibitory control were related to the ability of children to regulate their emotions. Moreover, motor and emotional inhibitory control measured in stop-signal tasks and emotion reappraisal tasks were related in adults (Tabibnia et al., 2011).»

References:

Carlson, S. M., Wang, T. S. (2007). Inhibitory control and emotion regulation in preschool children. Cognitive Development, 22(4), 489-510.

Chamberlain, S. R., Sahakian, B. J. (2007). The neuropsychiatry of impulsivity. Current opinion in psychiatry, 20(3), 255-261.

Cohen, J. R., Lieberman, M. D. (2010). The common neural basis of exerting self-control in multiple domains.Self control in society, mind, and brain, 9, 141-162.

Nigg, J. T. (2001). Is ADHD a disinhibitory disorder?. Psychological bulletin127(5), 571.

Palmwood, E. N., Krompinger, J. W., Simons, R. F. (2017). Electrophysiological indicators of inhibitory control deficits in depression. Biological psychology, 130, 1-10.

Tabibnia, G., Monterosso, J. R., Baicy, K., Aron, A. R., Poldrack, R. A., Chakrapani, S., ... & London, E. D. (2011). Different forms of self-control share a neurocognitive substrate. Journal of Neuroscience, 31(13), 4805-4810.

Authors also should explain why they decided to study children but not adults.

Response:

We decided to study children, because with age number of stressful events is increased. Also, as it has been shown that carriers of S allele more predisposed to disturbances in emotional regulation. The presence of S allele increases the risk for posttraumatic stress and major depression/anxiety spectrum disorders (Cervilla et al., 2006; Minelli et al., 2011; Xie et al., 2012). According to Minelli et al. (2000) development or presence of emotional disturbances could lead to contradictions on behavioral and brain activity results in genetic studies. Also, effect of 5-HTTLPR in children is less studied than in adults.

We added the following text to the manuscript (introduction section):

«It has been shown that carriers of S allele were more predisposed to disturbances in emotional regulation, the risk of posttraumatic stress, depression and anxiety disorders (Cervilla et al., 2006; Minelli et al., 2011; Xie et al., 2012). According to Minelli et al. (2011), development or presence of emotional disturbances in S allele carriers could lead to inaccurate data on behavioral and brain activity measures gathered from studies involving genetic polymorphisms. It can be assumed that the risk of developing psychopathologies increases with age due to the accumulation of stress and an increase in the number of stressful situations. An advantage of this study was that it would be performed on mentally and physically healthy children.»

References:

Cervilla, J. A., Rivera, M., Molina, E., Torres‐González, F., Bellón, J. A., Moreno, B., ... Nazareth, I. (2006). The 5‐HTTLPR s/s genotype at the serotonin transporter gene (SLC6A4) increases the risk for depression in a large cohort of primary care attendees: The PREDICT‐gene study. American Journal of Medical Genetics Part B: Neuropsychiatric Genetics, 141(8), 912-917.

Minelli, A., Bonvicini, C., Scassellati, C., Sartori, R., Gennarelli, M. The influence of psychiatric screening in healthy populations selection: a new study and meta-analysis of functional 5-HTTLPR and rs25531 polymorphisms and anxiety-related personality traits. BMC psychiatry, 2011, 11(1), 50.

Xie, P., Kranzler, H. R., Farrer, L., Gelernter, J. (2012). Serotonin transporter 5‐HTTLPR genotype moderates the effects of childhood adversity on posttraumatic stress disorder risk: A replication study. American Journal of Medical Genetics Part B: Neuropsychiatric Genetics, 159(6), 644-652.

Not always references to the relevant literature are provided. For instance:

"The serotonin transporter has a great importance in serotonergic neuromodulation, which is strongly linked to the emotion regulation".

Response: Thank you for the comments. We referred to Cools, R., Roberts, A. C., Robbins, T. W. (2008). Serotoninergic regulation of emotional and behavioural control processes. Trends in cognitive sciences, 12(1), 31-40.

"In studies of inhibition of motor reactions the stop-signal paradigm was well established".

Response: We referred to Verbruggen, F., Logan, G. D. (2008). Response inhibition in the stop-signal paradigm. Trends in cognitive sciences, 12(11), 418-424.

Methods

The authors should specify how they motivated children to perform the task of getting the maximum score.

The description of the experimental task is not always unambiguous. Providing the figure with the scheme of the probe with timing and recording details will substantially improve the manuscript. Some information regarding the task is missed: the total number of trials, the way of informing subjects about their performance (deduction and receiving "additional point").

Response: The Stop Sign task was implemented as a computer game, and most children like to play games. In addition, children from the same class participated in the study during the same time period and tried to get more points than their classmates.

Also, we provided following instruction in the text of article.

Prior to the start of the task, the following instruction appeared on the screen. «During the game, a rabbit or a tiger will be presented on the screen. You need to choose the right food - carrot for the rabbit (button K) and meat for the tiger (button D) and have time to press before the animal disappears. If a “STOP” signal appears on the animal, then nothing should be pressed. If you correctly feed the animal, you will receive a point, and in case of a wrong response (wrong food choice or if action was carried out after the "STOP" signal), the point will be deducted».

We added information about of number of trials.

«There were 160 trials. The first 30 trials without stop-signals were used in the training session. The practice session consisted of a pseudorandom mix of 96 (approximately 96%) no-stop and 34 (approximately 26%) stop trials.»

It is impossible to present a scheme of the probe with time interval because of time intervals were calculated individually.

The authors should provide an example of how the time interval between the target stimulus and stop-stimulus was calculated.

Response: We added the following sentences: «This can be represented as a formula - Mean RT in training section*0,1 or 0,2 or 0,7 or 0,8). For example, if mean RT in training section is 600 ms, then four variants of time interval between the target stimulus and stop-stimulus (600*0.1; 600*0.2; 600*0.7; 600*0,8) would be 60 ms, 120 ms, 420 ms and 480 ms respectively».

The package used for statistical analysis should be specified.

Response: We used the SPSS package.

We added the following sentence: «Statistical analysis was performed using the SPSS software package».

Results

Results section suffers from several drawbacks mainly associated with some degree of disagreement with the Methods section. Presented results revealed in many variants of analysis which wasn't introduced in the methods. More specifically, the authors should justify a number of issues:

The usage of particular subscales for psychological assessment. For what purposes they were used.; The usage of different ANOVAs; The selection of the 7-17 Hz frequency range of EEG activity, especially taking into account that alpha and beta demonstrated oppositely directed effects;

Without of abovementioned explanation, it is unclear how the performed types of analysis were motivated. Not all of these results were discussed in the rest of the text.

Thus the "results" section should be aligned with "methods" in terms of methods used and the analysis applied.

Response:

we rewrote "results" section according to the comments of the reviewers.

We used subscales of SDQ and ICID questionaires. We were more interested in scales of neurotisicm, hyperactivity-inattention, emotional and conduct problems. There are the studies where genotype group could differ on these trait or problems (Gizer et al., 2009; Minelli et al., 2011; Naughton et al., 2000; Papousek et al., 2013; van der Meer et al., 2014; Vaswani et al., 2003)

Such differences could influene to results of the polymorphism studies. In our study genotype groups did not differ on these scales. If there are differences in these subscales in genotype groups we would not be sure that the behavioral and EEG results revealed in LL, LS and SS groups were caused by effect of polymorphism of 5-HTTLPR.

We put before the revealed results the description of the analysis applied.

References:

Gizer IR, Ficks C, Waldman ID (2009) Candidate gene studies ADHD: a meta-analytic review. Hum Genet 126:51–90.

Minelli, A., Bonvicini, C., Scassellati, C., Sartori, R., Gennarelli, M. The influence of psychiatric screening in healthy populations selection: a new study and meta-analysis of functional 5-HTTLPR and rs25531 polymorphisms and anxiety-related personality traits. BMC psychiatry, 2011, 11(1), 50.

Naughton M, Mulrooney JB, Leonard BE (2000) A review of the role of serotonin receptors in psychiatric disorders. Hum Psychopharmacol 15:397–415.

Papousek, I., Reiser, E. M., Schulter, G., Fink, A., Holmes, E. A., Niederstätter, H., ... Weiss, E. M. (2013). Serotonin transporter genotype (5-HTTLPR) and electrocortical responses indicating the sensitivity to negative emotional cues. Emotion, 13(6), 1173.

van der Meer D, Hartman CA, Richards J, Bralten JB, Franke B, Oosterlaan J, Heslenfeld DJ, Faraone SV, Buitelaar JK, Hoekstra PJ (2014) The serotonin transporter gene polymorphism 5-HTTLPR moderates the effects of stress on attention-deficit/hyperactivity disorder. J Child Psychol Psychiatry 55:1363–1371.

Vaswani M, Linda FK, Ramesh S (2003) Role of selective serotonin reuptake inhibitors in psychiatric disorders: a comprehensive review. Prog Neuropsychopharmacol Biol Psychiatry 27:85–102.

Discussion

The discussion regarding the impact of age to the observed phenomenon, i.e., comparison between similar effects demonstrated for the adults, will strengthen the manuscript.

Response: We did not find papers written in English where researchers investigated the oscillatory dynamics during inhibition of movement in LL, LS and SS groups in adults. We found only one paper in which oscillatory dynamics were investigated in LL, LS and SS groups of adults but unfortunately this paper was written in Russian language and we did not include it because it would not be available for most researchers who do not understand Russian.

The link between emotional regulation, cognitive control, and inhibitory control of motor activity should be thoroughly discussed.

Response: In the revised manuscript we discussed only inhibitory control of motor activity. In the introduction section we added text about link between emotional regulation and inhibitory motor control. The motor control is one of the aspects of cognitive control.

Finally, it is not clear why the authors did not try to correlate the psychometric assessment and revealed changes in EEG activity.

Response:

We did not try to correlate the psychometric assessment and oscillatory responses. It was not an aim of this study. We were aimed to investigate the effects of the 5-HTTLPR polymorphism on oscillatory dynamics during successful inhibition of motor reaction. In this study we used subscales of SDQ and ICID questionaires (mainly scales of neurotisicm, hyperactivity-inattention, emotional and conduct problems) for to exclude their possible impact in the different genotype groups. There are the studies in which SS genotype group differed from LL and LS on these trait or problems (Gizer et al., 2009; Minelli et al., 2011; Naughton et al., 2000; Papousek et al., 2013; van der Meer et al., 2014; Vaswani et al., 2003) and according to Minelli et al. (2011) such higher scores or severity could impact to the results in SS group. Fortunately, in our study LL, LS and SS groups did not differ on these traits or problems.

The text of the manuscript contains several lexical mistakes.

Response: Native English speaker revised the paper.

Round 2

Reviewer 1 Report

After the first round of review, the paper is improved; however, there still several minor issues that should be fixed before the endorsement.

Major points

English is improved; however, it could be further improved as there are still several mistakes. Therefore, please, ask your Native English speaker to revise the text again.

Besides, you have to state in the limitations that you did not compute the stop-signal reaction time (SSRT), relying on the existing literature. As your sample is unique, you might incur some degree of inaccuracy.

INTRODUCTION

Lines 72-76. Rearrange the sentence as follows: ‘It has been shown that different kinds of inhibitory control (motor and affective inhibitory control) share a common psychobiological substrate [14,18]. In addition, impaired motor inhibitory control was associated with depression and impulsivity [15,16]. For instance, children with ADHD took longer to inhibit their initial motor response, and reported disturbances in emotional regulation [17]’. Lines 88-91. Rearrange the sentence as follows:’ The stop signal paradigm is well established paradigm for the study of pre-planned movement [24]. This task consists of a pseudorandom mix of GO and stop trials. In go-trials subjects have to move quickly and accurately in response to the presentation of the target stimulus. Less frequently, a stop-signal is presented during the reaction time (RT) and the participant is instructed to inhibit the pending action’ Please DO NOT LABEL ‘Go trials’ with ‘target trials’. Similarly, DO NOT MISNAME ‘reaction times’ with ‘time of reaction’. Change those expressions throughout the manuscript.

METHODS

Lines 151-152 Rearrange the sentence as follows: ‘The response keys were not counterbalanced across subjects’. Lines 161-166 Rearrange the sentence as follows: ‘Similarly, in this study we computed the time interval between the onset of the target and stop-signal stimuli (stop signal delay, SSD) for each participant based on the average RT measured in the first 30 trials. The SSD was 10%, 20%, 70% and 80% to the average RT. Thus, for instance, if the mean RT in the training section was 600 ms….’ Lines 189 -190 Rearrange the sentence as follows: ‘…the time it takes to to the stop signal, or stop-signal reaction time (SSRT) was about 200 ms’ Line 191 Change the text as follows ‘In this study we did not compute the SSRT but we rely on measures taken from the existing literature [33,35,36 maybe add Federico and Mirabella (2014) Exp Brain Res. 232:1293-307], thus we considered the first 200 ms after the presentation of the stop signal as the SSRT’.

RESULTS

Table 1 Go RT must be in ‘ms’ not in %. Please, correct. Personally, I would try to show in all figures significant differences via a ‘*’

DISCUSSION

Line 296 Change to ‘subjects’ ability’ Line 304 Change to ‘The RT’

Author Response

We would like to thank the reviewer for attention and scrupulous consideration our manuscript. Thank you for your detailed comments and suggestions. I have revised the manuscript according your suggestions. Most of changes were highlighted in red colour. Only, one point I could not make. I could not represent significant differences between 3 genotype groups via a *. Unfortunately, I had no experience in showing differences via a * in figures.